# Practical Suggestions for Assessing Rabbit Haemorrhagic Disease Virus 2 Risk to Endangered Native Lagomorphs in North America and Southern Africa

**DOI:** 10.3390/v16081299

**Published:** 2024-08-15

**Authors:** Brian Cooke

**Affiliations:** Foundation for Rabbit Free Australia, P.O. Box 145, Collinswood, Adelaide, SA 5081, Australia; briancooke.rfa@gmail.com

**Keywords:** RHDV, RHDV2, European rabbit, lagomorphs, North America, South Africa, Europe, ecology, disease risk

## Abstract

A new form of the rabbit haemorrhagic disease virus, RHDV2, first observed in European rabbits, has spread widely among different species of hares in Europe, jackrabbits and cottontails in North America, and hares in southern Africa. However, only limited surveillance studies have been undertaken so far. It is suggested that methods developed for controlling the disease in farmed rabbits in Europe and studying the efficacy of RHDV as a biological control agent in Australia could facilitate epidemiological research on those recently affected lagomorph species. This would enable the assessment of the risk of RHDV2 to native lagomorphs, including endangered species, and the determination of the main host species of RHDV2. Because RHDV2 has not spread equally through all lagomorph species, epidemiological studies could give insights into factors important for determining host susceptibility.

## 1. Introduction

Rabbit haemorrhagic disease (RHD) was first described in China among domestic rabbits (*Oryctolagus cuniculus*) imported from East Germany in 1984 [1]. It is a severe and usually fatal disease involving major liver damage and disseminated intravascular coagulation [2]. The virus rapidly spread through wild and domestic rabbits in Europe, causing serious economic damage to the rabbit meat and fur industries until an effective vaccine was introduced. It spread to domestic rabbits in many other countries, including Korea, Mexico, Egypt, and Lebanon [3], and was deliberately introduced into Australia in 1995 and to New Zealand in 1997 for the biological control of introduced European rabbits, which are major economic and ecological pests in both countries [4,5].

The causative virus, rabbit haemorrhagic disease virus (RHDV), has been described and found to only infect European rabbits, while the closely related European brown hare syndrome virus (EBHSV) causes a similar disease in European brown hares (*Lepus europaeus*). EBHSV occasionally spreads to other hare species such as the mountain hare (*L. timidus*) and the Corsican hare (*L. corsicanus*), as well as wild eastern cottontails (*Sylvilagus floridanus*) introduced in Italy [6,7,8,9,10]. An experimental challenge of North American black-tailed jack rabbits (*Lepus californicus*) and endangered Mexican volcano rabbits (*Romerolagus diazi*) with RHDV did not result in disease [11].

RHD and EBHSV, which infect rabbits and hares (lagomorphs), have since been classified as lagoviruses, a genus within the family Caliciviridae, and as additional related viruses have been found, a standard system of nomenclature has subsequently been recommended [12]. RHDV is now frequently referred to as GI.1 and EBHSV is GII.1, while related non-pathogenic rabbit caliciviruses which circulate in both domestic and wild rabbits [13,14] and the hare calicivirus from captive-bred and wild hares [15] are referred to as RCV/GI.3,4 and HaCV/GII.2, respectively.

In 2010, a new lagovirus, referred to as RHDV2/GI.2, emerged in France and rapidly spread worldwide [16,17,18]. Initially less virulent than RHDV, the virus increased in virulence and spread much more readily among young rabbits than RHDV [19], but it also spread to several different species of hares in Europe. These included the Cape and the Italian hares (*L. capensis* and *L. corsicanus*), the Iberian hare (*Lepus granatensis*), and the mountain hare (*L. timidus*) and its Irish subspecies (*L. timidus hibernicus*) [20,21,22,23,24].

RHDV2 also appeared in the USA and Canada in domestic rabbits. Whereas earlier outbreaks of RHDV had been isolated and infrequent, in 2020 it began spreading among black-tailed jackrabbits and antelope jackrabbits (*Lepus alleni*), desert cottontails (*Sylvilagus audubonii*), mountain cottontails (*S. nuttallii*), brush rabbits (*S. bachmani*), and to a lesser extent eastern cottontails (*S. floridanus*). Outbreaks centred in the western USA and northern Mexico caused conspicuous mortality in several lagomorph species [25]. More recently, RHDV2 has been reported in South Africa, where it apparently infects Cape hares (*L. capensis*), scrub hares (*L. saxatilis*), and red rock hares (*Pronolagus* spp.). The extent of its spread among other species, such as the African savanna hare (*L. victoriae*), remains uncertain due to difficulties in sampling hares and the accurate identification of different hare and rabbit species (B. Schumann pers. comm.).

The spread of RHDV2 among so many different lagomorph species has become a major wildlife conservation issue. In Mexico, which is home to fourteen lagomorph species, there are two endangered species, the volcano rabbit (*Romerolagus diazi*) and the Davis Mountains cottontail (*Sylvilagus robustus*), that could potentially be infected. Additional species of concern in the USA include California’s endangered riparian brush rabbit (*Sylvilagus bachmani riparius*) [26] and the pygmy rabbit (*Brachylagus idahoensis*). Populations of the pygmy rabbit in the Colombia Basin of Washington State are considered at risk of extinction even without the added threat of a new lethal disease [27]. Likewise, in South Africa, work is underway to see if RHDV2 is spreading through populations of the rare riverine rabbit (*Bunolagus monticularis*). The three species of red rock hares (*Pronolagus rupestris*, *P. crassicaudatus*, and *P. randensis*) may also be at risk.

In the western USA, RHDV2 is now endemic in at least eleven states, including Texas, Colorado, New Mexico, Arizona, Utah, California, Montana, and Wyoming, although there are indications that the virus spread is slowing [28] and lagomorph populations appear to be recovering from the initial disease outbreak [29]. In Mexico, RHDV2 has affected cottontails and jackrabbits, mainly in the states of Baja California, Baja California Sur, Chihuahua, Coahuila, Durango, and Sonora.

The spread of RHDV2 in North America has significant economic implications. In the USA, the domestic rabbit industry is estimated to be worth between USD 2.2 and 2.3 billion annually, most of which reflects the value of pet supplies for the care of over 6.7 million pet rabbits in approximately 2.9 million households [30]. The Mexican Government previously made an enormous effort to eradicate RHDV from domestic rabbits [11], and that was possible only because there was no reservoir of disease in wild cottontails or jackrabbits at the time. Unfortunately, with RHDV2 now well established, there is a constant risk of spread of RHDV2 between domestic and wild lagomorphs. This necessitates a more costly approach to disease control by regularly vaccinating domestic rabbits to protect the rabbit meat and fur industries and household pets [31].

Shapiro et al. [32] note that, in North America, surveillance standards that apply to domestic animal health are difficult to apply to wildlife health surveillance, which is typically based on the ‘convenience sampling’ of dead or visibly sick animals (also referred to as passive or scanning surveillance). Agencies responsible for wildlife health surveillance are hampered by several factors, including incomplete ecological knowledge of wildlife populations at risk, limited ability to determine the true disease status of wildlife populations, pathogen persistence or transmission outside vertebrate hosts (i.e., vector-borne pathogens), insufficient diagnostic facilities and staff, regulatory restrictions, and funding constraints.

Because of the passive surveillance approach to RHDV2 thus far [32], no detailed epidemiological studies on affected cottontail and jackrabbit populations in North America have been published, and little is known about the behaviour of RHDV2 in recently infected lagomorph species. However, it can be surmised from recent observations that the disease has not severely compromised any species of lagomorph to the point that they have become critically endangered. Furthermore, although eastern cottontails have been experimentally infected with RHDV2 [33], the virus seems less likely to spread naturally in this host than among desert cottontails or mountain cottontails. The latter two species are more like European rabbits in ecology and behaviour than other cottontails [34].

Closer studies are essential because the lack of epidemiological information is holding back a better understanding of this new disease. Such studies would be a worthwhile investment because even small-scale serological studies that are guided by our current knowledge of the behaviour of RHDV in European rabbits are likely to be highly informative. Indeed, the methods developed jointly by Australian and Italian researchers from 1995 onwards to actively monitor the effectiveness of RHDV as a biological control agent [35,36] could be applied to assess the impact of RHDV2 on other lagomorph populations. Even more importantly, theory relating to the epidemiology of RHD [37] and the coevolution of the virus and its hosts [38,39,40,41] is critical for understanding how RHDV2 might evolve and affect lagomorph populations in the long term.

Using some practical examples, it is shown here how insights into the impact of RHDV2 outbreaks might be quickly obtained from relatively small field samples of rabbits and hares at relatively low cost. This would enable the current passive surveillance approach to the spread of RHDV2 to be upgraded to active surveillance for the better management of lagomorphs in North America and southern Africa.

From a wider evolutionary perspective, comparative studies of RHDV2 epidemiology in different lagomorph species, especially comparisons between species in which RHDV2 has spread rapidly and those that seem refractive, would provide valuable insights into factors that determine the emergence of new diseases and the subsequent co-evolution of viruses and their new hosts [42]. Such studies have wider implications for understanding newly emerged viruses in general.

## 2. Review of Key Factors Influencing RHD Epidemiology and Rabbit Survival

### 2.1. Rabbit Susceptibility

Although RHD caused very high mortality among susceptible sub-adult and adult rabbits when it first appeared, it was noted that very young rabbits were resilient to the disease [3]. Young, unweaned rabbits often survived the disease even though their mothers died [3]. Those young could be successfully raised by immune foster mothers in the case of domestic rabbits, but usually died of starvation in the wild.

Nonetheless, this juvenile resilience gradually declined, and rabbits became highly susceptible to the disease by 12 weeks of age [3,4]. This contrasts with myxomatosis, where young rabbits are more likely to die from the disease than older rabbits [4]. Consequently, it is important to understand the complex interactions between age and the risk of mortality to gauge the effectiveness of RHDV as a biological control agent and to assess threats to domestic farmed rabbits and wild populations in countries such as Spain and Portugal, where rabbits are part of the natural fauna [43].

Very young rabbits less than 2 weeks old can be infected with RHDV, but the virus does not spread beyond localized foci in the liver. Ruvoën-Clouet et al. [44] suggested that this might be because RHDV binds to ABH blood group antigens of rabbit epithelial cells of the upper respiratory tract, but the tissues of very young rabbits are almost devoid of A and H type 2 antigens, leading to the very weak binding of virus particles and low infection rates. Nonetheless, when ABH antigens are expressed more strongly, at 12 weeks of age, rabbits become highly susceptible to infection and severe disease.

This explanation is inadequate, however, because it does not explain how hepatocytes, which do not express ABH antigens, become infected. Furthermore, the resilience of young rabbits clearly involves an immune response because, if treated with the immunosuppressant methylprednisolone acetate (MPA), rabbit kittens develop fulminant hepatitis and die within 3 days [45]. Indeed, Neave et al. [46] have shown that young rabbits have a robust immune response to RHDV but a lesser response to RHDV2, suggesting that their immune response is mediated by the viruses.

As young rabbits grow older, the risk of severe infection increases. Robinson et al. [47] experimentally analysed and quantified these changes. They developed a mathematical model to demonstrate that young rabbits below 2 weeks of age were naturally very resilient to disease and that maternal antibodies added further protection against acute disease depending on the antibody titre of the mother (Figure 1).

Maternal antibodies in the serum of rabbit kittens are predominantly of the IgG isotype, and can be readily identified and quantified using appropriate ELISA techniques [35]. The IgG titres of maternal antibodies in near-term (gestational age 28–29 days) rabbit embryos from shot rabbits are like those of their mothers (Cooke unpublished), and so IgG antibody titres in new-born wild rabbits must be similar. But as young rabbits increase in age, maternal antibody titres decline, and their protective effects are lost. 

Based on the maternal antibody titres of wild rabbit kittens trapped and individually tagged, then re-trapped at short intervals in the field, the half-life of maternal antibodies is 6.65 ± 0.50 days (Cooke unpublished), similar to the estimated half-life of maternal antibodies against the rabbit myxoma virus of about 7 days [48]. Note that maternal antibodies not only break down over time but are also diluted as young rabbits grow, so a half-life in this case is determined by a combination of factors (i.e., it is not like the decay of radioactive materials).

Among wild rabbits, few breeding female rabbits have reciprocal IgG titres above 10,240 [35,49], and with a maternal antibody half-life of only 6.65 days, antibody titres approach the IgG iso-ELISA ‘cut-off’ of a 1 in 10 dilution between 9 and 10 weeks (Figure 2). Maternal antibodies in wild rabbits are generally not detectable and have little protective role in rabbits beyond 70 days (10 weeks) old [35,47]. 

Importantly, Baratelli et al. [50] demonstrated very similar patterns of maternal antibody decline in rabbit kittens borne by RHDV2-vaccinated does. Vaccination does not produce such high titres as direct exposure to RHDV2, but maternally derived antibodies against RHDV2 were still detectable until young rabbits were about 28 days old. Furthermore, the cross-fostering of young rabbit kittens born to unvaccinated mothers onto vaccinated ones [50,51] showed that not only were maternal antibodies transferred across the placenta, but they could also be transferred to suckling rabbits in milk. This potentially explains why the half-life of maternal antibodies in the rabbits in some published experiments appeared longer than the 6.65 days measured directly in young wild rabbits that had left the maternal nest and were no longer suckling. Importantly, it also implies that antibodies in rabbit milk should impede infection by neutralizing the virus particles before they can bind to mucosal cells. 

Even so, despite such natural defences against infection, young rabbits can still become infected with the virus, although this does not always happen easily. Rabbit kittens in the wild remain in the maternal nest until 3 weeks old then seldom range far from burrows within a defended territory, meaning they have little direct contact with older rabbits likely to be infected with RHDV. However, it is still possible for susceptible sub-adult rabbits in the social group to become infected and spread the disease to younger rabbits or for carrion-eating flies to spread the virus by feeding on dead rabbits and defecating on pasture vegetation [52].

A small experiment (Cooke unpublished) confirmed that maternal antibodies do not protect against RHDV infection. Newly captured juvenile wild rabbits between 6 and 10 weeks old were kept for 2 days alongside an RHDV-infected individual before being transferred to separate cages. Some of those young rabbits had maternal antibodies while others were seronegative, having been borne by seronegative does. Notably, 6 out of 10 rabbits with maternal antibodies and 13 out of 23 rabbits without maternal antibodies contracted RHD, indicating that both groups were equally susceptible to infection under these circumstances (χ^2^ = 0.03; *p* > 0.5). However, only 2/13 (17%) of the juvenile rabbits without maternal antibodies survived, whereas all 6 rabbits with antibodies survived (χ^2^ = 13.38; *p* < 0.001).

These experimental results are consistent with the expectations derived from Figure 1 above, where among 8-week-old rabbits, only 17% would be expected to survive if born with no maternal antibodies, but 92% would be expected to survive if born to a doe with a reciprocal IgG antibody titre of 10,240.

Hall et al. [53] have since experimentally confirmed that antibodies raised against RHDV2 protect against disease, but not infection. Thus, when rabbits are infected while they still retain maternal antibodies, a high proportion should survive. In the field, they would then be recruited into the adult breeding population.

### 2.2. Rate of Virus Spread and Survival from Disease

In the experimental situation described above, the spread of RHDV among the young rabbits was determined by close contact with an infected rabbit. In wild populations, other factors are likely to be important too, such as the immunological structure of the population, climate, time of host breeding, and the abundance of vectors. In disease models, the combined effect of all such variables determines the rate at which susceptible individuals in a population acquire an infection and is sometimes termed the force of infection [54,55]. If the disease spreads very rapidly it follows that susceptible rabbits must be younger when first infected [37], and this has significant implications where many rabbits are below 12 weeks old when infected and survive because of age-related resilience and the persistence of maternal antibodies.

From field data collected during live-trapping field studies [35,49] in the years immediately after RHDV spread in Australia, it was determined that RHDV spread relatively slowly and rabbits rarely became infected with RHDV until they were more than 12 weeks old (about 850 g body weight), as by then they had lost all their natural protection. 

Data from live-trapped wild rabbits, marked with numbered ear tags and serologically tested for antibodies to both RHDV and MYXV, were used to provide an estimate of the severity of mortality among unprotected rabbits. As Table 1 shows, rabbits can be assigned to different antibody categories, e.g., seropositive for both RHDV and MYXV, seropositive for one or the other, or seronegative for both, and their survival can be followed as both diseases complete their spread through the population. Such data can then be analysed, assuming that for rabbits with antibodies to both diseases the probability of survival is determined by predation or other natural causes alone (S_n_), whereas for rabbits with antibodies to RHDV, survival is determined by the product of the probabilities of survival from myxomatosis and survival from natural mortality factors (S_m_ × S_n_). For rabbits that lack antibodies to either virus, their survival rate is the product of the probabilities of surviving both diseases and other natural mortality factors (S_r_ × S_m_ × S_n_). From rabbit survival data, the probability of S_n_ is 0.75, and S_m_ and S_r_ are estimated as 0.46 and 0.14, respectively. 

The data in Table 1 were also analysed using analysis of variance and analysis of deviance to enable the uncertainty of survival estimates to be calculated (Table 2). Nonetheless, these results are constrained by small sample sizes. To achieve better long-term estimates of the survival of adult rabbits immune to both diseases, the problem of small data sets might be partially overcome by setting more precise values of S_n_ as priors in Markov Chain Monte Carlo (MCMC) simulations using a Bayesian approach to estimate parameters [56].

Importantly, however, despite limits to the data, this example indicates that the probability of survival from RHD in the field is very low, like the survival rates recorded experimentally [47]. In 1997, soon after RHD first spread across Australia, it heavily reduced rabbit abundance, especially when combined with myxomatosis. Similar analyses of combined live-trapping and serological data would be of value in assessing the impact of RHDV2 where it has spread into new lagomorph hosts.

When conditions were suitable for RHDV to spread rapidly, however, there was a different outcome. Rabbit kittens and juveniles less than 12 weeks old were then infected, and many survived. Raw field data from live-trapped rabbits monitored at the Gum Creek field site in South Australia [35,49] can be set out in a similar way to that shown in Figure 2 to make this clearer, the only difference being that the body weight (g) of young rabbits is used in place of age. This is justifiable because, up to a body weight of 1000 g, young wild rabbits grow at a steady rate of about 10 g/day on average [57].

The first data set (Figure 3a) was collected in 1996 when RHDV spread slowly, while the second (Figure 3b) was collected from the same locality in 2000 when RHDV spread rapidly, and most rabbits were quickly infected with RHDV irrespective of age. 

The difference between the two years shown in Figure 3a,b can be assessed in terms of the relative numbers of seronegative rabbits (25 vs. 5) as well as the number of recently infected but recovering rabbits (2 vs. 11). Rabbits were more abundant in the year when RHDV spread slowly, as indicated by the number of individual rabbits caught over four nights of trapping with 40 cage traps set (68 vs. 39). Thus, the force of infection is not a simple function of rabbit density or the proportion of rabbits susceptible to infection, as commonly assumed in ‘mass action’ epidemiological models [58,59], and additional factors such as vector abundance, rabbit behaviour (fighting, mating), and host–virus evolution probably play a role. 

There are more subtle differences to be observed as well. When the force of infection was high, the relationship between maternal antibodies and body weight declined in a predictable way, as shown in Figure 2. For example, because the half-life of maternal antibodies is 6.65 ± 0.50 days and, in that time, the average young rabbit’s body weight increases at 10 g per day [57], i.e., by 66.5 g, the slope of the relationship between maternal antibody titres and body weight was expected to be approximately −log_10_2/66.5 = −0.0045 ± 0.0004

Indeed, when young rabbits were rapidly infected, irrespective of age and maternal antibody titres, the slope of the fitted regression was −0.0048, close to the expected value as determined from maternal antibody half-life. However, when RHDV spread slowly, young rabbits born with low maternal antibody titres lost those antibodies and disappeared from the ‘maternal antibody’ group first, whereas those born with high titres remained recognizable for a longer period. This distorted the apparent relationship between maternal antibody titres and body weight, making it appear as though maternal antibodies in general were persisting longer and as shown in Figure 3a, the slope of the fitted regression was −0.0031.

Although the rate of spread of RHDV was not closely linked to rabbit density in the limited data presented here, it is generally considered to be highest where host populations are most abundant [58,59]. This was also assumed in the model proposed by Calvete [37], who noted that, in Spain, RHD had little effect in areas where rabbits were abundant, whereas it caused the disappearance of rabbits in areas where they were least abundant. Calvete argued that, where rabbits were abundant, density-dependent infectivity would cause rabbits to contract RHD earlier in life, enabling enough recruitment to maintain populations. By contrast, in areas where rabbits were scarce and the virus spread slowly, many rabbits would only be infected once they were over 12 weeks old. The consequent high mortality rate then resulted in population attrition.

### 2.3. Rabbit and Virus Coevolution

After RHDV first spread among wild rabbits in Australia in late 1995, outbreaks of RHDV followed a consistent pattern. Between 1996 and about 2000, it was commonly observed that RHDV spread among young adult rabbits during autumn (March–May) in the Southern Hemisphere [38,39,40,49,60]. Those rabbits had been born in late spring (October–November) of the previous year and had not been infected by RHDV because of the low rates of infection. Furthermore, they had not been challenged over the summer months when RHDV spread poorly [59,60], and by the following autumn they were well over 12 weeks old and highly susceptible to the disease. With the onset of the rabbit breeding season in late autumn and heightened social contact due to territorial defence and mating, most of those rabbits contracted RHD and died. RHD then broke out again about five months later in the following spring (October–November) among young born early in the breeding season (from April onwards) that had lost their juvenile resilience and maternal antibody protection. It only occasionally spread among younger late-born rabbits, however.

Mutze et al. [38,39,40,41] subsequently presented evidence that this initial pattern steadily changed over time. They showed that outbreaks of RHD became largely confined to the spring months, and that outbreaks began earlier and were more prolonged. The age of rabbits that died decreased, and fewer adult rabbits were involved in outbreaks. Mutze et al. [40] considered that ‘Increased infection and virus shedding in juvenile rabbits offers the most plausible explanation for those epidemiological changes; the disease is now increasingly transmitted through populations of kittens, starting before young-of-the-year reach adult size and persisting late in the breeding season, so that most rabbits are challenged in their year of birth. These changes have increased juvenile mortality due to RHD but reduced total mortality across all age groups, because age-specific mortality rates are lower in young rabbits than in older rabbits’.

The epidemiological changes that Mutze et al. [40] describe are consistent with Figure 3a,b, which show that, in 1996, immediately after the initial spread of RHDV into wild rabbits, few became infected before they were 12 weeks old, whereas by 2000, rabbits were more commonly infected as youngsters while they still retained some juvenile resilience and maternal antibodies.

Interestingly, the data imply that when RHDV was first released in Australia as a new pathogen, there may have been an initial mismatch between resistance to infection and virus virulence. Young rabbits were either too resilient, or more likely, virus virulence was not sufficiently attuned to infect the rabbits without initiating a strong immune response [46]. Coevolutionary adjustment would then have quickly followed because rabbits infected as kittens survived better, and virus variants which could infect young rabbits spread before those which could not. Elsworth et al. [61,62] provide experimental evidence of the evolution of rabbit resistance to infection and the likely adjustment of virus virulence in response.

Mutze et al. [41] showed that, after 2002, rabbit populations in South Australia began to increase once again, presumably because of these observed epidemiological adjustments. Wild rabbits also regained some of their former abundance following the initial impact of RHD in Spain. Much of this increase occurred between the early 2000s and 2013 [63] and, because it preceded the arrival and broad establishment of RHDV2 in Spain [18], it was attributed to changes in infrastructure such as roads and railway lines, which acted as potential corridors along which resistant rabbits could spread. Nonetheless, coevolutionary adjustment between RHDV virulence and rabbit resistance cannot be excluded given other evidence that viruses and hosts are co-evolving [61,62].

When RHDV2 arrived in Australia in 2015, it rapidly displaced RHDV as the most common virus [64]. Presumably, it had a selective advantage because it could more easily infect young rabbits and so spread earlier in the year [63], but this also implies that RHDV2 probably had less impact on populations than RHDV. In Spain, annual hunting bag records indicate that the spread of RHDV2 had a far smaller impact compared with the earlier spread of RHDV [65].

## 3. Assessing Risk to Other Lagomorphs

From what is known of RHDV in wild European rabbits, it can be concluded that the impact of RHDV2 on native lagomorphs in Europe, North America, and South Africa will depend heavily upon the rate of infection. The initial spread of RHDV2 through any naïve lagomorph population would kill many susceptible adults and young, but in the following year it would spread mainly among the next cohort of young and, if it spread readily, many of those young would survive because they would carry maternal antibodies contributed by mothers that survived the initial outbreak, and possibly because of natural juvenile resilience too.

Thus, for lagomorph populations where RHDV2 has recently spread, a simple initial step towards understanding the risk from RHDV2 could be taken by shooting or cage-trapping 50 or more rabbits or hares from the field in late spring when RHDV2 is spreading and collecting and analysing serum samples from them to define their serological status (i.e., seronegative, maternal antibodies, recently infected, etc.).

For the proper definition of RHDV2-induced antibody titres, specific reagents have been produced at the World Organization for Animal Health (WOAH) Reference Laboratory for RHD, and ELISA methods capable of providing a specific response have been developed and validated. RHDV2 monoclonal antibodies (Mabs) have been produced and characterized and used to develop a competition ELISA (cELISA) for RHDV2. Similarly, methods for the assay of anti-isotype immunoglobulins (IgG, IgA and IgM) for RHDV2 have been adapted and developed. Further details on these methods are published in the *Manual of Diagnostic Tests and Vaccines for Terrestrial Animals* (https://www.woah.org/fileadmin/Home/fr/Health_standards/tahm/3.07.02_RHD.pdf) (accessed on 14 August 2024).

The cELISA RHDV2 method is commercially available (https://www.izsler.it/chi-siamo/per-chi-e-con-chi-lavoriamo/centri-di-referenza/internazionali/oie-reference-laboratory-for-rabbit-haemorrhagic-disease/diagnostic-reagents-and-kits/ (accessed on 14 August 2024)), whereas assays for antibody isotypes are performed upon request by the WOAH Laboratory. Some RHDV/RHDV2 Mabs are also available for scientific purposes on request through the Italian Biobank of Veterinary Resources (http://www.ibvr.org (accessed on 14 August 2024)).

Following appropriate ELISA titrations, and a check that there are no interacting viruses such as RCVs and HaCVs to confuse the picture, the IgG titre of each animal and its RHDV2 serological status could then be plotted against some broad indicators of age, like body weight, as shown in Figure 3a,b, allowing useful conclusions to be drawn.

If the pattern resembled that shown in Figure 3a, with few recovering animals, numerous susceptible sub-adult animals, and a relatively low slope in the relationship between maternal antibody titres and body weight, then it would indicate that young animals are not being infected until they are sub-adults. The mortality of sub-adults would then be high and recruitment into the breeding population low. On the other hand, if there are numerous recently recovered animals, few seronegative sub-adults, and a sharper apparent decline in maternal antibody titres with increasing host body weight, as in Figure 3b, then the impact of disease will be less because young rabbits are being infected but survive because they have maternal antibodies and possibly age-related resilience. Recruitment into the breeding population would then be higher, and the risk from RHDV2 lower.

Because RHDV2 is better able to infect young rabbits than RHDV, despite age-related resilience [66,67], it enables more young rabbits to survive than RHDV, and this probably explains why RHDV2 has had a relatively small impact on European rabbit populations in Spain and Portugal [65] and why, in many areas around Valencia and Aragon, rabbit populations have recently increased substantially, as judged by the increase in numbers shot by hunters [63,65].

For endangered lagomorphs, collecting samples of serum from large numbers of individual animals may not be possible. Nonetheless, in the case of endangered riparian brush rabbits in California, where some hundreds have been trapped and vaccinated to protect them from RHDV2 [68,69], enough samples of sera could have been obtained to not only provide insights into the risk of RHDV2 to the population, but to provide information on the effectiveness of vaccination in raising antibodies in rabbits. As suggested in Table 1 and Table 2, the recapture of brush rabbits could potentially enable a comparison of the survival rates of seronegative rabbits that were vaccinated and those which had survived natural infection with RHDV2.

This approach to disease surveillance is far more useful than passive surveillance because it is based on an understanding of the mechanism that enables lagomorph populations to overcome outbreaks of severe disease, rather than simply detecting the presence of disease.

Data of this kind would go well beyond simply obtaining a better understanding of RHDV2, and would provide a unique opportunity to document the spread of new diseases in wildlife populations and to consider the many environmental variables enabling the establishment of such diseases and their capacity to spread. Such studies would also provide a unique opportunity to make progress in unravelling many of the factors that determine the species specificity of RHDV2, i.e., whether they are genetic, behavioural, or climate-related.

The present situation, with RHDV2 newly spreading among several different lagomorph species on three continents, presents an unprecedented opportunity for studying evolving virus–host relationships that should not be wasted. Much has been learned from the release of the myxoma virus into European rabbit populations in both Australia and Europe, enabling the coevolution of rabbit resistance and virus virulence to be closely studied in duplicated natural experiments on two continents [70,71]. A closer study of RHDV2 could yield similar insights.

## Figures and Tables

**Figure 1 viruses-16-01299-f001:**
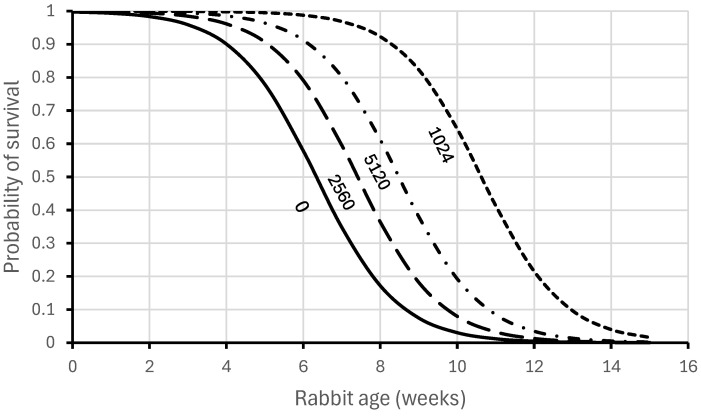
Relationship between surviving experimental RHDV challenge and age of young rabbits (redrawn from data in Robinson et al. [47]). Reciprocal IgG antibody titres of mothers are shown below each curve.

**Figure 2 viruses-16-01299-f002:**
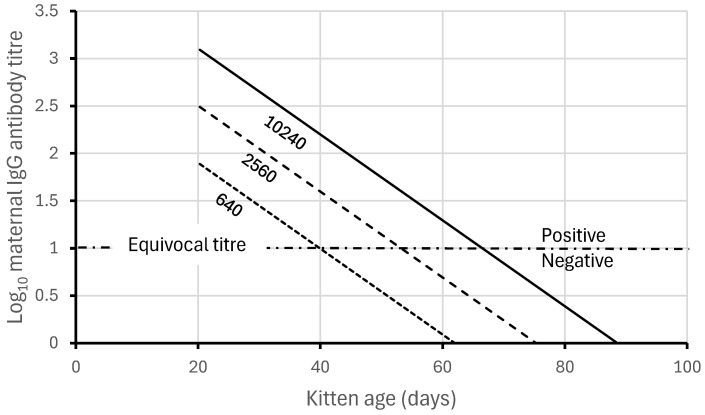
Knowledge of average reciprocal IgG titres of maternal antibodies (generally less than 10,240) and their rate of loss or decay (half-life 6.65 days) in young rabbits after weaning at 20 days. The sensitivity limits of the IgG iso-ELISA explain why maternal antibodies are rarely detected in juvenile rabbits for more than 70 days (10 weeks old and about 700 g body weight). The log scale for maternal antibody titres linearizes the antibody decay curves.

**Figure 3 viruses-16-01299-f003:**
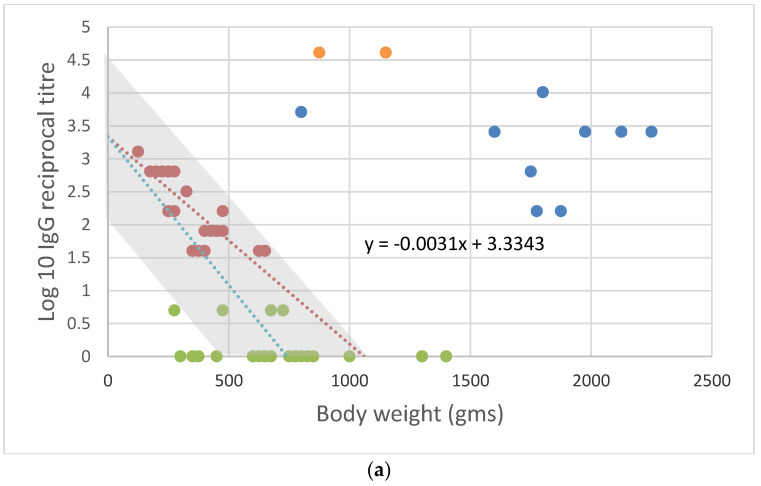
(**a**) Antibody pattern in late 1996, a year after RHDV first spread in Australia when the virus only spread slowly through the whole rabbit population. Reciprocal IgG titres (log scale) are plotted against the live weight of rabbits (nearest 25 g). Green—questionably positive and seronegative rabbits; dark red—rabbits with detectable maternal antibodies (IgG only); orange—recently infected rabbits (IgM titre greater than IgG titre); mid-blue—rabbits with IgG and IgA antibodies indicating prior infection and general immunity. Dark-red broken line—regression fitted to data from rabbits with detectable maternal antibodies. Broken light-blue line—maternal antibody decay anticipated from half-life of maternal antibodies. Shaded area—range of maternal antibody titres expected based on maternal antibody half-life and the range of titres observed in breeding adult rabbits. (**b**) Antibody pattern in late 2000, after RHDV had been circulating for five years and the RHDV had begun spreading rapidly through the whole population each spring. Symbols are as specified for (**a**). Note that there are very few seronegative rabbits (green data points) and many recently infected surviving rabbits (orange data points). The slope of the fitted regression is −0.0048, matching the slope of −0.0046 ± 0.0004 expected from maternal antibody half-life.

**Table 1 viruses-16-01299-t001:** Example of method used for estimating mortality caused by RHDV and myxoma virus (MYXV) based on the number of rabbits in each antibody class captured and individually marked with a numbered ear tag in late summer, 1997, and recaptured the following winter when both RHD and myxomatosis had spread to all rabbits (i.e., every rabbit recaptured in June 1997 was seropositive for antibodies raised against both viruses).

			Antibody Categories in February		
RHDV/MYXV Antibodies	+/+	+/−	−/+	−/−	Total
Rabbits captured February ‘97	4	23	1	20	48
Rabbits in each category recaptured June ‘97	3	8	0	1	12

**Table 2 viruses-16-01299-t002:** Probability of survival from disease estimated from the field data above for rabbits with and without antibodies to RHDV and MYXV. Although sample sizes within immune categories were small, analysis of deviance indicated high deviance ratios giving support for the model fit.

	Four-Month Survival Probability ± s.e.		
	With Antibodies	Without Antibodies	Probability of Survival from Each Disease
RHDV	0.392 ± 0.060	0.054 ± 0.034	0.14
MYXV	0.470 ± 0.119	0.220 ± 0.039	0.47

## Data Availability

Unpublished data associated with antibody titres in rabbit embryos and live-trapping studies, as shown in figures, are available from the author on reasonable request.

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
