# Peer review of "Practical Suggestions for Assessing Rabbit Haemorrhagic Disease Virus 2 Risk to Endangered Native Lagomorphs in North America and Southern Africa"

_viruses, 2024, doi:10.3390/v16081299_

Round 1

Reviewer 1 Report

Comments and Suggestions for Authors

Practical suggestions for assessing RHDV2 risk to endangered 2 native lagomorphs in North America and Southern Africa

Introduction of RHDV/RHDV2 in Australian rabbit populations have provided rich opportunities to study the epidemiological risk of new viral disease in a geographic area, and the review described and recommend active methods that involve capturing live wild rabbits to trace the viral spread in different lagomorph species by showing some practical examples used in Australia. The manuscript would benefit by including survey examples on RHDV/RHDV2 in wild rabbits in Europe and adding more detailed information on the figures.

Comments:

Table 1(a): The table legend and the table are very confusingly described. Are the same rabbits captured and recaptured in 1997? The legend explains that all re-captured surviving rabbits had RHDV and MYXV antibodies – does it mean that the same rabbits in each category were re-captured and counted? It would be helpful to revise the table and the legend for clarification.

Table 1(b): how the calculation was done?

The rabbit capture studies in Figure 2 and 3, table 1 – have they been published elsewhere and presented here in a modified format? Please add references or add more descriptions on the studies (locations and other relevant information)

Line 337-346: This paragraph is somewhat vague: the author mentioned that lower infection in young rabbits in 1996 may be the result of viral force of infection and resistance to infection, which may be due to lower viral virulence or rabbits being more resistant. What was the prevalence of RHDV at the time of sample collection in Fig 3? Can the lower infection in young rabbits in 1996 be explained by lower prevalence of RHDV at that time? Is there published evidence that viral virulence has changed during 1996 and 2000? Is it due to genetically more susceptible rabbits dying off in early spreading years? Adding this information would be helpful to readers who do not have background information on RHDV in Australia.

Line 405-406: It is unclear what the sentence (Ramsay et al..) means.

Author Response

Reviewer 1 

Comment 1. Introduction of RHDV/RHDV2 in Australian rabbit populations have provided rich opportunities to study the epidemiological risk of new viral disease in a geographic area, and the review described and recommend active methods that involve capturing live wild rabbits to trace the viral spread in different lagomorph species by showing some practical examples used in Australia. The manuscript would benefit by including survey examples on RHDV/RHDV2 in wild rabbits in Europe and adding more detailed information on the figures.

Response: Thank you for the suggestions. Where relevant I have now referred to studies of RHDV2 in Europe e.g. Santoro et al (2023) line 401, reference [65]

Comment 2. Table 1(a): The table legend and the table are very confusingly described. Are the same rabbits captured and recaptured in 1997? The legend explains that all re-captured surviving rabbits had RHDV and MYXV antibodies – does it mean that the same rabbits in each category were re-captured and counted? It would be helpful to revise the table and the legend for clarification.

Response: Yes, that is precisely what was done. Rabbits were individually earmarked, so it was known which ones had survived. I have revised the table, legend and added more text to make it clearer for readers to follow and understand (see Lines 243-256 in revised text).

Comment 3. Table 1(b): how the calculation was done?

Response: The data can be analysed using different approaches ranging from simple rough estimates of survival to more complex methods to estimate uncertainty about estimates. I have added text to explain how this can be done (see Lines 257-263 of revised text).

Comment 4. The rabbit capture studies in Figure 2 and 3, table 1 – have they been published elsewhere and presented here in a modified format? Please add references or add more descriptions on the studies (locations and other relevant information)

Response: Thank you for pointing out that more information is needed here. I have rewritten the text relating to Figure 2 including the caption to indicate where the data originate. The figure itself is simply a reworking of available data described in the text to build a conceptual model that helps the reader to interpret Figures 3a and 3b.

Comment 5. Line 337-346: This paragraph is somewhat vague: the author mentioned that lower infection in young rabbits in 1996 may be the result of viral force of infection and resistance to infection, which may be due to lower viral virulence or rabbits being more resistant. What was the prevalence of RHDV at the time of sample collection in Fig 3? Can the lower infection in young rabbits in 1996 be explained by lower prevalence of RHDV at that time? Is there published evidence that viral virulence has changed during 1996 and 2000? Is it due to genetically more susceptible rabbits dying off in early spreading years? Adding this information would be helpful to readers who do not have background information on RHDV in Australia.

Response: From Figs. 3a and 3b RHD was less prevalent in the spring of 1996 than in 2000 (i.e. there were fewer infected rabbits) but that is not the point. Differences in prevalence (detected by sampling at the same time in two different years) reflect the rate of spread rather than being a condition enabling spread. There is good, published evidence by Mutze et al. (1998; 2008; 2014; 2015) of epidemiological changes over the period 1996 – 2000 which imply changes in the way the virus and host interacted, and these are reviewed in this manuscript. There are also data showing that virus and host are probably coevolving (Elsworth et al 2012; 2014) and I have added a note to the text including those references ([61, 62] Lines 384-386). A new sub-heading in the manuscript ‘Rabbit and virus coevolution’ enables this information to be more easily found.

Comment 6. Line 405-406: It is unclear what the sentence (Ramsay et al.) means.

Response: To avoid adding a paragraph of explanation I have deleted the reference to Ramsey et al. 2019.

Reviewer 2 Report

Comments and Suggestions for Authors

Rabbit hemorrhagic disease has been known as a threat to domestic and wild rabbit. A new lagovirus, RHDV2, emerged in France and has rapidly spread throughout the world. The author reviewed practical suggestions for assessing RHDV2 risk. However, the manuscript includes reviews on not only RDHV2 but also RDHV. First, the title of this review should be reconsidered to reflect the contents of the manuscript precisely. The manuscript seems to be well prepared but some revisions are required as follows.

L20-134: It is recommended to divide the introduction into an actual introduction portion and an epidemiological information portion. The author refers to the subsequent section as a “Review of available epidemiological information”. Despite this, the section covers resilience to RHDV in young rabbit, maternal antibodies and protection, and other areas. In contrast, the author reviewed the epidemiological information in lines 21-134. It is suggested that the “Introduction” section be categorized and reworked.

L135: As indicated above, the title of this part should be reconsidered.

L237-257: Citations of references based on these descriptions are required.

L276-301: Citations of references based on these descriptions are required.

L313-325: What data was utilized by the author in writing this paragraph?  Display supporting evidence for this paragraph. Reference 59 was published in 1998 and included data from 1995 to 1996. L314-316: the author described data from 1996 to 2000. Cite reference(s) for this description.

L363: It is unclear what “the timing of disease outbreaks” and “the force of infection” mean. Clarify descriptions.

Author Response

Referee 2

Comment 1. Rabbit hemorrhagic disease has been known as a threat to domestic and wild rabbit. A new lagovirus, RHDV2, emerged in France and has rapidly spread throughout the world. The author reviewed practical suggestions for assessing RHDV2 risk. However, the manuscript includes reviews on not only RDHV2 but also RDHV. First, the title of this review should be reconsidered to reflect the contents of the manuscript precisely. The manuscript seems to be well prepared, but some revisions are required as follows.

Response: Thanks for pointing out where readers might be confused by discussion about RHDV and RHDV2. Nonetheless,  I have reconsidered the title of the manuscript, trying to constructively include RHDV. However, all my efforts detracted from the main aim of providing practical suggestions for assessing RHDV2 impact – so I have not changed the title. The brief abstract and key words also add information not covered by the title.

Comment 2. L20-134: It is recommended to divide the introduction into an actual introduction portion and an epidemiological information portion. The author refers to the subsequent section as a “Review of available epidemiological information”. Despite this, the section covers resilience to RHDV in young rabbit, maternal antibodies and protection, and other areas. In contrast, the author reviewed the epidemiological information in lines 21-134. It is suggested that the “Introduction” section be categorized and reworked.

Response: I agree with the reviewer’s comment. Accordingly, I have changed the heading ‘2. Review of available epidemiological information’ to ‘2. Review of key factors influencing RHDV epidemiology and rabbit survival’ and removed a section from the introduction and incorporated it in this second review section.

Comment 3. L135: As indicated above, the title of this part should be reconsidered.

Response: The title of this part has been changed to ‘2. Review of key factors influencing RHDV epidemiology and rabbit survival’ and subheadings have been inserted e.g., ‘2.1 Rabbit susceptibility’, ‘2.2 Rate of virus spread’, ‘2.3 Rabbit and virus coevolution.’

Comment 4. L237-257: Citations of references based on these descriptions are required.

Response: This section has been rewritten as follows.

From field data collected during live-trapping field studies [35, 49] in the years immediately after RHDV spread in Australia, RHDV spread relatively slowly and rabbits rarely became infected with RHDV until they were more than 12 weeks old (about 850 g body weight), and by then, they had lost all their natural protection (Lines 239 - 242). Line 243 - 256 give details of how such data can be analysed.

Comment 5. L276-301: Citations of references based on these descriptions are required.

Response: The section of text here is a fuller explanation of the significance of data given in Figures 3a and 3b. The figures present raw field data provided to help someone wishing to assess the risk of RHDV2 to a wild lagomorph population to interpret field data. As such, it has not previously been published in that form, but has been drawn from data collected for earlier studies. I have added a note to that effect and listed references indicating the studies from which the examples were drawn (see lines 282-285).

Comment 6. L313-325: What data was utilized by the author in writing this paragraph?  Display supporting evidence for this paragraph. Reference 59 was published in 1998 and included data from 1995 to 1996. L314-316: the author described data from 1996 to 2000. Cite reference(s) for this description.

Response: I thank the reviewer for pointing out that more refeences are needed. References from Mutze et al. (1998; 2008; 2014; 2015) and Cooke et al (2002; 2018) have been added [38-41, 49, 56] to show that monitoring of epidemiology continued well beyond 1996.

Comment 7. L363: It is unclear what “the timing of disease outbreaks” and “the force of infection” mean. Clarify descriptions.

Response: I agree with the reviewer. Force of infection is not an easy term to understand and is measured in terms of the rate spread of the virus anyway. Therefore, I have rewritten the text in terms of the rate at which RHDV spreads through a population (see lines 224-242). I have also deleted ‘the timing of disease outbreaks’ as this is largely taken up in the concept that the virus spreads through each cohort of susceptible young produced each year. Earlier in the manuscript I have referred to this as the ‘time of host breeding’ (see Line 232 of altered manuscript )

Round 2

Reviewer 1 Report

Comments and Suggestions for Authors

The revised manuscript now contains more detailed information. 

Reviewer 2 Report

Comments and Suggestions for Authors

The author uses only an abbreviation, RHDV2, in the title. It is better to add the full name of RHDV2, rabbit hemorrhagic disease virus 2, in the title.